# Comparing physician associates and foundation year two doctors-in-training undertaking emergency medicine consultations in England: a mixed-methods study of processes and outcomes

Mary Halter [1], Vari Drennan [1], Chao Wang,[1] Carly Wheeler,[2] Heather Gage,[3] Laura Nice,[4] Simon de Lusignan [5], Jonathan Gabe,[6] Sally Brearley,[1] James Ennis,[7] Phil Begg,[8] Jim Parle[4]

**Correspondence to**
Mary Halter;
m.halter@sgul.kingston.ac.uk

## ABSTRACT

**Objectives** To compare the contribution of physician associates to the processes and outcomes of emergency medicine consultations with that of foundation year two doctors-in-training.

**Design** Mixed-methods study: retrospective chart review using 4 months' anonymised clinical record data of all patients seen by physician associates or foundation year two doctors-in-training in 2016; review of a subsample of 40 records for clinical adequacy; semi-structured interviews with staff and patients; observations of physician associates.

**Setting** Three emergency departments in England.

**Participants** The records of 8816 patients attended by 6 physician associates and 40 foundation year two doctors-in-training; of these n=3197 had the primary outcome recorded (n=1129 physician associates, n=2068 doctor); 14 clinicians and managers and 6 patients or relatives for interview; 5 physician associates for observation.

**Primary and secondary outcome measures** The primary outcome was unplanned re-attendance at the same emergency department within 7 days. Secondary outcomes: consultation processes, clinical adequacy of care, and staff and patient experience.

**Results** Re-attendances within 7 days (n=194 (6.1%)) showed no difference between physician associates and foundation year two doctors-in-training (OR 0.87, 95% CI 0.61 to 1.24, p=0.437). If seen by a physician associate, patients were more likely receive an X-ray investigation (OR 2.10, 95% CI 1.72 to 4.24), p<0.001), after adjustment for patient characteristics, triage severity of condition and statistically significant clinician intraclass correlation. Clinical reviewers found almost all patients' charts clinically adequate. Physician associates were evaluated as assessing patients in a similar way to foundation year two doctors-in-training and providing continuity in the team. Patients were positive about the care they had received from a physician associate, but had poor understanding of the role.

**Conclusions** Physician associates in emergency departments in England treated patients with a range of conditions safely, and at a similar level to foundation

## Strengths and limitations of this study

► This study provides a well-powered quantitative comparative analysis of the documented processes and outcomes of patient care by physician associates and foundation year two doctors-in-training in three emergency departments in different parts of England.

► We believe this to be the first empirical study of the outcomes of care provided by UK-trained physician associates in the emergency department, and the first internationally to include interview and observation data.

► Patients' views have not been previously reported for physician associates in this setting.

► The low sensitivity of the emergency department triage system to identify conditions other than the most serious was a problem and impaired the study's ability to describe case mix fully.

year two doctors-in-training, providing clinical operational efficiencies.

## INTRODUCTION

Healthcare systems internationally are challenged to ensure good patient outcomes, within financial constraints, as well as to attend to the work life of the workforce.[1] Health workforce shortages, particularly of doctors, are resulting in the development of advanced clinical practitioners or non-physician clinicians (NPCs), such as nurse practitioners (NPs) and physician assistants/associates (PAs) in many countries.[2] Numerous countries are experiencing rising patient demand for emergency services and concomitant shortages of doctors in emergency medicine.[3–7] This situation has led to the development of NPC roles in emergency

departments (EDs) in many countries such as the USA,[8] Australia,[9] Canada[10] and the UK.[11] In the USA, 25% (n=14360) of all emergency medicine clinicians are NPCs, and 68% of these are PAs.[8]

PAs are trained in the medical model to take histories, diagnose illness, develop management plans and prescribe medications as agreed with their supervising physician. PAs have a 50-year history in the USA and are a developing part of the workforce in some other countries such as Canada, the Netherlands and Germany.[12] The PA workforce is growing in the UK (where they are known as physician associates). In 2018, there was an estimated 600 qualified PAs with approximately 1000 graduating each year since then.[11] Their employment specialties include EDs,[11] where they are deployed in both the minor and the major illness or injury sections.[8]

Descriptive observations have been published concerning the positive contributions by US-trained PAs employed in EDs in the UK,[13] Australia and New Zealand,[14] and by UK-trained PAs in England.[15] Unlike in the USA, PAs in these other countries cannot prescribe medicines or order ionising radiation. PAs in North American EDs are reported to be well accepted by other staff and patients, and reliable in assessing certain medical complaints and performing procedures.[16] No difference is reported between patients attended by a PA and those attended by a doctor for wound infection rates, or rate of revisit within 72 hours to a paediatric ED; but studies find less consistency in practice when analysing prescribing patterns, length of stay and wait times of physicians, PAs and NPs in the ED.[17] There is relatively little research evidence on their clinical effectiveness,[17] little quantitative evidence on outcomes from outside of the USA and no qualitative evidence of how PAs deliver care in the ED. In this context, our goal was to investigate the contribution of PAs to the processes and outcomes of emergency medicine consultations compared with that of foundation year two (FY2) doctors-in-training in EDs in English hospitals.

## METHODS
### Study design
We conducted a pragmatic, mixed-methods convergence study in which we compare and contrast and simultaneously interpret quantitative and qualitative data[18] in three EDs in England, with three components. We undertook a quantitative observational retrospective chart review of patient consultations by PAs compared with FY2 doctors-in-training; and qualitatively we directly observed PAs' practice; and we conducted semi-structured interviews with members of the staff team. Our planned prospective study of patient records with a linked patient satisfaction and outcomes survey had to be revised to a pragmatic retrospective chart review due to practicalities within the participating National Health Service (NHS) organisations in the period of the study.

### Population and sampling
Three consultant-led, 24 hours EDs with full resuscitation facilities ('type one') participated. Two EDs had annual attendances in the range of 100000–120000 adult and paediatric patients and the third in the range of 170000–190000. One was an university hospital; two were district general hospitals. The hospitals had been recruited as part of a larger study investigating the work and contribution of PAs between 2016 and 2017.[19] We selected FY2 doctors-in-training as the comparator for PAs, as PAs are offered as part of a solution to junior medical workforce shortages[7] and the most junior doctors working in the UK ED at the time were FY2s.

### Selection of participants, measurements and outcomes
Our primary outcome was unplanned re-attendance at the same ED within 7 days—one of the NHS clinical quality indicators for EDs in England.[20] Our secondary outcomes were: consultation processes (length of time in the ED, use of X-ray, prescriptions and referrals); clinical adequacy of care, referrals and planned follow-up; and patient experience.

### Chart review
For a 16-week period (the standard duration of ED placement for FY2 doctors-in-training in the UK), we obtained anonymised, routinely collected electronic records of all patients attended by a PA or FY2 doctor-in-training, provided in Microsoft Excel by the hospital information teams in each trust, using queries based on staff job role, dates and requested data items. Hospital staff extracted additional data items (online supplementary material 1)—age, sex, acuity (as categorised by the Manchester triage score[21]), X-ray orders, diagnosis, prescription issued, admission, area treated, overall time in the ED (from check in to discharge, in minutes) and re-consultation within 7 days (the primary outcome). No data linkage was required. The researchers did not have access to the original dataset and so could not identify if any patients appeared more than once in the dataset, and further data cleaning could not be performed.

We calculated a sample size for the primary outcome based on rate of 18.3% (the highest of two rates for nurse practitioners substituting for physicians (at 28 days)).[22 23] Aiming to find a relative difference of 50%, in a non-inferiority hypothesis, we required 284 patients in each group (calculation from Stata V.11.1 software) to compare 18.3% to 27.4% unplanned re-consultations, with conventional 80% power at 5% significance. We included an extra 20 to allow for adjustment for case mix, requiring a minimum of 304 patients in total in each group to achieve the said power. As 28-day data could not be collected, we went on to use the 7-day re-attendance rate, with its national average of 7.4% (range 2.4%–21.7%) for unplanned re-attendance at the same ED in England for all patients.

Two of the participating EDs also agreed to take part in the analysis of clinical adequacy of documented care

in every 10th case from the chart review sample (n=40), with equal numbers of cases seen by PA and FY2 doctors-in-training, and using the full anonymised clinical record. We recruited two specialty registrars (doctors in their 6th year of emergency medicine training), one PA lecturer (with 20 years ED experience) and one emergency medicine consultant (with 17 years experience at consultant level) from outside the three study hospitals to review these records. All four clinicians independently recorded their judgement as to the clinical adequacy of care for each record using the categories of medical history, examination, request for radiography, treatment plan and decision, advice given and follow-up. Their assessments were blinded to the type of professional attending the patient and to each other's assessment, using a proforma (online supplementary material 2) based on published studies.[22 23] As the senior clinician, we accepted the decision of the consultant in cases of disagreement.

## Observation

This element drew on the ethnographic tradition used in many health service research studies.[24] We invited all PAs working in the ED in our three study hospitals to participate (n=6). Five PAs volunteered and gave written informed consent to be observed. One of three researchers (CWh, LN, MH) observed each PA for two or three pre-arranged sessions, of varying lengths, on weekdays in periods between 08:00 and 22:00 hours, following a broad guide (online supplementary material 3). Researchers made notes on context, relationships and activities following this guide. We judged data saturation to have been reached with individual PAs when the processes of care observed did not differ significantly from previous observations. During the observation period, PAs asked for patient assent to the researcher's presence. Researchers reflected on the observations, discussing them in pairs.

## Interview

Semi-structured interviews[25] were undertaken with a purposive sample of managerial, medical and nursing ED staff who volunteered after receiving information about the study from the researcher during observation periods and/or via their site manager. We also opportunistically interviewed patients and/or their relative who were being seen by a PA in the ED, identified and invited to participate during observation periods, once they had been assessed and treated by the PA but before discharge from the ED. We used tailored topic guides (online supplementary material 4) to explore interviewees' perceptions of the PA role and its impact on service organisation, role boundaries, patient experience, patient outcomes and activities and attitudes of other staff. We digitally recorded interviews or took notes if the participant preferred. Recordings were transcribed verbatim and anonymised.

## Analysis
### Chart review

The characteristics of patients treated by PAs and FY2 doctors-in-training were compared using $\chi^2$ tests. We carried out a logistic regression to examine whether the primary and binary secondary outcomes differed between PAs and FY2 doctors-in-training, while adjusting for confounding factors—patient age, sex and triage score. Since patients seen by the same clinician are likely to be correlated, we calculated intraclass correlation coefficients (ICC) for each outcome and report results using a random-effects model if the ICC is statistically significant. We report ORs, their CIs and two-tailed p values. For length of stay, a linear regression was used for data transformed to logarithm scale to reduce heteroscedasticity and reflect the fact that the value of length of stay is positive. To account for unobserved heterogeneity, the unobserved component is modelled as a latent variable in a latent class linear model. The assessment of clinical adequacy is reported using descriptive statistics, sensitivity and specificity of the judgement of whether the record was that of a PA or FY2 doctor-in-training and Fleiss kappa for inter-rater agreement, calculated for each of the four components of the assessment and per response.

### Qualitative

Our methods for the analysis of observation data drew on methods to identify ethnographic vignettes.[26] We employed thematic analysis[27] of all-specialty interview data for the wider study. Both are described in full elsewhere.[19] For the subsequent specialty-specific analysis, we re-read all ED observation data and interview transcripts to identify all data related to the primary and secondary outcomes, and which both confirmed or disconfirmed findings.

### Mixed methods

Following the separate quantitative and qualitative analyses, we (MH and VD, in consultation with all authors) merged[18] the quantitative and qualitative datasets by presenting the quantitative results by study outcomes and following these with qualitative data findings (themes and/or excerpts or quotes) that confirmed or disconfirmed the quantitative results.

### Patient and public involvement

The patientand public voice was important to this study and informed the design, conduct, analysis, interpretation and final reporting. We brought the views of our public and patient representative forum for a previous study on physician associates into the research questions and design of the study. These were views such as how do patients understand this new role. Sally Brearley, as a public voice representative, was a co-applicant and member of the research team. The study advisory group had two public voice members who were reimbursed for their time, following NIHR INVOLVE guidance. Two patient and public voice groups were formed: one

**Table 1** Characteristics of chart review sample

| Characteristic | PA (n=2381) N (%) | FY2 doctor (n=6435) N (%) | Total (n=8816) N (%) | P value |
|---|---|---|---|---|
| **Age band (years)** | | | | |
| 0–20 | 300 (13.0%) | 656 (10.3%) | 956 (11.0%) | 0.002 |
| 21–40 | 543 (23.5%) | 1493 (23.5%) | 2036 (23.5%) | |
| 41–60 | 530 (22.9%) | 1406 (22.1%) | 1936 (22.3%) | |
| 61–80 | 551 (23.8%) | 1596 (25.1%) | 2147 (24.7%) | |
| 81 and over | 390 (16.9%) | 1212 (19.0%) | 1602 (18.5%) | |
| **Sex** | | | | |
| Male | 1132 (47.5%) | 2933 (45.6%) | 4065 (46.1%) | 0.102 |
| Female | 1249 (52.5%) | 3501 (54.4%) | 4750 (53.9%) | |
| **Manchester triage score** | | | | |
| 1 Immediate | 10 (0.6%) | 3 (0.1%) | 13 (0.2%) | <0.001 |
| 2 Very urgent | 163 (9.3%) | 565 (11.1%) | 728 (10.6%) | |
| 3 Urgent | 770 (43.8%) | 2841 (55.7%) | 3611 (52.6%) | |
| 4 Standard | 811 (46.1%) | 1681 (32.9%) | 2492 (36.3%) | |
| 5 Non-urgent | 5 (0.3%) | 12 (0.2%) | 17 (0.2%) | |
| **ED area treated in** | | | | |
| Minor | 369 (20.1%) | 275 (6.8%) | 644 (10.9%) | <0.001 |
| Major | 1266 (68.8%) | 3601 (88.4%) | 4867 (82.3%) | |
| Resuscitation | 2 (0.1%) | 4 (0.1%) | 6 (0.1%) | |
| Paediatrics | 181 (9.8%) | 174 (4.3%) | 355 (6.0%) | |
| Clinical decision unit or primary care | 21 (1.1%) | 20 (0.5%) | 41 (0.7%) | |

FY2, foundation year two; PA, physician associates.

in London and the other in the West Midlands and members reimbursed as per NIHR Involve guidelines. The patient and public voice groups informed the design of the research tools such as topic guides and participant information sheets, developed coding frameworks and analysed interview transcripts, and participated in the overall synthesis of findings. Sally Brearley continues to be involved in the dissemination of the study.

## RESULTS
### Characteristics of chart review subjects
In the 16-week period studied, 8816 patients seen by 6 PAs (n=2890) or 40 FY2 doctors-in-training (n=5926) were identified; some secondary outcomes were available for all cases. For 3197 of these patient episodes (n=1129 by the 6 PAs and n=2068 by 22 FY2 doctors-in-training), the primary outcome was collected at site for the research team. Characteristics of the patients are shown in table 1. PAs saw a lower proportion of patients categorised on triage into the urgent category than FY2 doctors-in-training.

In interview, the type of patient seen, patient throughput and role of PAs and FY2 doctors-in-training were described as similar:

They're [the PAs] pretty much equal to ……a senior FY2 doctor in training level. As a consultant we feel comfort because we know [PA name 1] can work in majors, she can clear [majors] pretty much…… And [PA name 2],… can clear paeds minors… Participant 150 emergency medicine consultant

However, more than one participant tentatively suggested that PAs saw the less acutely unwell patients:

So my understanding is like they're [the PAs] equivalent to, I would put it like a certain level of like a junior physician……I wouldn't say they would be at registrar level……I'd put them somewhere in between. You know a…lot better than like a newly qualified physician because they've got the skills and stuff, so in that gap of what I would say equivalent to maybe like a second to four years post qualified doctor. Participant 144 registrar

### Characteristics of interview and observation participants
The staff interviewed included four PAs, two managers, five nurses and three senior doctors; six patients and/ or their relatives were also interviewed, spread across the three sites. We observed four PAs, at three sites; we do

**Table 2** Re-attendance at the same ED within 7 days

| Re-attendance at the same ED within 7 days | PA (n=1129) | FY2 doctor (n=2068) | Total (n=3197) | Unadjusted OR (95% CI) and p value (PA relative to FY2 doctor-in-training) in rate of re-attendance | Adjusted OR (95% CI) and p value (PA relative to FY2 doctor-in-training) in rate of re-attendancee* |
|---|---|---|---|---|---|
| No | 1066 (94.4%) | 1937 (93.7%) | 3003 (93.9%) | 0.87 (0.64 to 1.19) p=0.388 | 0.87 (0.61 to 1.24) p=0.437 |
| Yes | 63 (5.6%) | 131 (6.3%) | 194 (6.1%) | | |
| Unknown | 1251 | 4368 | 5619 | – | – |

*Adjustment made for triage score (as a measure of acuity), age band, sex, admission, X-ray and site; no adjustment was made for clustering as the ICC by individual staff member on outcome was small (0.008) and statistically insignificant (p=0.236).
ED, emergency department; FY2, foundation year two; ICC, intraclass correlation coefficient; PA, physician associates.

not report further demographic details due to concerns about anonymity in a small population.

### The primary outcome: rate of return to the ED within 7 days

Re-attendance within 7 days was found following 6.1% (n=194) of the 3197 index visits for which these data were available. The high rate of unknown data is accounted by one site where these data were not captured in the electronic dataset and were only retrieved manually for a random sample (n=205) for the purposes of this study. After adjustment for confounding, no statistically significant difference was found for cases seen by PAs or FY2 doctors-in-training (table 2).

### Secondary outcome: consultation processes

No differences were found between patients attended by PAs or by FY2 doctors-in-training in: whether prescriptions were given, admission to hospital from the ED or if a discharge summary was completed. However, patients seen by a PA were more likely to have an X-ray performed in the ED (table 3), less likely to be admitted to hospital and to have a shorter length of stay in the ED (by 35 min), after adjustment for age, sex, acuity, whether admitted, X-ray taken and site, as well as for clustering by individual clinician, although no account was able to be taken of the staffing level.

**Table 3** Clinical process measures

| Clinical process measure | PA (n=2381) | FY2 doctor (n=6435) | Total (n=8816) | Unadjusted OR (95% CI) and p value (PA relative to FY2 doctor-in-training) in rate of re-attendance | Adjusted OR (95% CI) and p value (PA relative to FY2 doctor-in-training) in rate of re-attendancee* |
|---|---|---|---|---|---|
| **X-ray investigations performed** | | | | | |
| No | 559 (49.4%) | 1701 (82.3%) | 2260 (70.7%) | 4.76 (4.04 to 5.59) p<0.001 | 2.70 (1.72 to 4.24) p<0.001 |
| Yes | 572 (50.6%) | 366 (17.7%) | 938 (29.3%) | | |
| Unknown | 1250 | 4368 | 5618 | – | – |
| **Prescriptions given in the ED** | | | | | |
| No | 174 (58.0%) | 157 (51.8%) | 331 (54.9%) | 0.79 (0.56 to 1.07) p=0.127 | 1.35 (0.08 to 23.5) p=0.838 |
| Yes | 126 (42.0%) | 146 (48.2%) | 272 (45.1%) | | |
| Unknown | 2081 | 6132 | 8213 | | |
| **Admitted as an inpatient from the ED** | | | | | |
| No | 883 (78.2%) | 1436 (70.1%) | 2319 (73.0%) | 0.65 (0.55 to 0.77) p<0.001 | 0.78 (0.55 to 1.1) p=0.158 |
| Yes | 246 (21.8%) | 613 (29.9%) | 859 (27.0%) | | |
| Unknown | 1762 | 3876 | 5638 | | |
| **Discharge summary completed** | | | | | |
| No | 86 (42.4%) | 71 (34.6%) | 157 (38.5%) | 0.72 (0.48 to 1.08) p=0.109 | 1.57 (0.93 to 2.66) p=0.09 |
| Yes | 117 (57.6%) | 134 (65.4%) | 251 (61.5%) | | |
| Unknown | 2178 | 6230 | 8408 | | |

*Adjustment made for MTS (as a measure of acuity), age band, sex and site, and for clustering where the ICC (and p value) is significant: X-ray 0.04 (p<0.001), prescriptions 0.73 (p<0.001), admitted 0.02 (p=0.001), discharge summary <0.001 (p=0.498).
FY2, foundation year two; ICC, intraclass correlation coefficient; PA, physician associates.

We observed PAs being the first member of the medical team to carry out assessment of patients following triage to either the major, minor or paediatric areas of the ED. We noted that PAs saw patients independently, following a medical history taking and examination model, before reporting in person to the senior ED physician in the same way as nurse practitioners and FY2 doctors-in-training did.

PAs were differentiated from FY2 doctors-in-training by many of our interviewees for not being able to prescribe medications or order tests using ionising radiation. Some participants considered this to have a detrimental impact on PAs and patients:

> [prescribing] would make a massive difference for them as well and [for] patients because at the end of the day they're having to wait for the PAs to go talk through [with] the physicians what's going on and then probably see somebody else. Participant 118 nurse practitioner

However, PAs were observed taking on several roles in relation to prescriptions and X-ray orders, for example, suggesting medications to or charting the medication for a senior doctor to sign off:

> So when one of my PAs comes to me and says 'This patient has a temperature of 38, they're coughing up horrible green sputum and they're tachycardic and I listened to their chest and they've got crackles at the left base, can we order a chest x-ray and prescribe sepsis drugs for, you know, pneumonia?' I say 'Yes' and I sign it. With probably more confidence at this stage having had [number] PAs here for a year than I would with a junior physician in training on day two. And the irony of that is of course, the junior physician in training doesn't need to come and ask me, technically, they can prescribe themselves. Participant 21 emergency medicine consultant

PAs were also observed making referrals to medical and surgical teams outside of the ED, completing discharge summary information and carrying out procedures, most commonly cannulation, phlebotomy and suturing.

### Secondary outcome: clinical adequacy

Our reviewers found the chart documentation to have been 'appropriate' or 'with no errors or omissions that resulted in significant probability that the patient might be harmed' in 36/40 cases for all of the key consultation components (table 4). In the three records (two of FY2 doctors-in-training and one of a PA) judged as having errors or omissions at the level of a breach in normal guidelines and procedures that would have altered the patient's treatment, all reviewers agreed that a senior doctor review had occurred in one case; this was unclear in the other cases. Our observation data suggest that such a senior review was undertaken for all assessment and clinical decision making in the 'majors' sections

**Table 4** Chart reviewers' assessments of clinical adequacy

| PA or FY2 consultation record | | Judgement of appropriateness | | | | | | | | | | | | | | | | | | | | | | | |
| | | Medical history | | | | Examination | | | | Request for radiography* | | | | Treatment plan and decision | | | | Advice given | | | | Follow-up | | | |
| | | Appropriate | Error or omission | | | Appropriate | Error or omission | | | Appropriate | Error or omission | | | Appropriate | Error or omission | | | Appropriate | Error or omission | | | Appropriate | Error or omission | | |
| | | | Harm unlikely | Altered treatment | Harm | | Harm unlikely | Altered treatment | Harm | | Harm unlikely | Altered treatment | Harm | | Harm unlikely | Altered treatment | Harm | | Harm unlikely | Altered treatment | Harm | | Harm unlikely | Altered treatment | Harm |
|---|---|---|---|---|---|---|---|---|---|---|---|---|---|---|---|---|---|---|---|---|---|---|---|---|---|
| FY2 | n | 14 | 6 | 0 | 0 | 15 | 5 | 0 | 0 | 9 | 0 | 1 | 0 | 14 | 5 | 1 | 0 | 4 | 1 | 1 | 0 | 16 | 3 | 0 | 0 |
| | % | 70 | 37 | 0 | 0 | 75 | 25 | 0 | 0 | 45 | 0 | 5 | 0 | 70 | 25 | 5 | 0 | 20 | 5 | 5 | 0 | 80 | 15 | 0 | 0 |
| PA | n | 13 | 5 | 1 | 0 | 11 | 7 | 1 | 0 | 9 | 3 | 0 | 0 | 13 | 5 | 1 | 0 | 3 | 1 | 1 | 0 | 13 | 4 | 1 | 0 |
| | % | 65 | 25 | 5 | 0 | 55 | 35 | 5 | 0 | 45 | 15 | 0 | 0 | 65 | 25 | 5 | 0 | 15 | 5 | 5 | 0 | 65 | 20 | 5 | 0 |
| Not rated* | n | 1 | | | | 1 | | | | 18 | | | | 1 | | | | 29 | | | | 3 | | | |
| | % | 2.5 | | | | 2.5 | | | | 45 | | | | 2.5 | | | | 73 | | | | 7.5 | | | |
| Total | n | 27 | 11 | 1 | 0 | 26 | 12 | 1 | 0 | 18 | 3 | 1 | 0 | 27 | 10 | 2 | 0 | 7 | 2 | 2 | 0 | 29 | 7 | 1 | 0 |
| | % | 68 | 28 | 3 | 0 | 65 | 30 | 3 | 0 | 45 | 8 | 3 | 0 | 66 | 25 | 5 | 0 | 18 | 5 | 5 | 0 | 73 | 18 | 3 | 0 |
| Agreement (Fleiss kappa, combined) | | 0.01 | | | | 0.15 | | | | 0.26 | | | | 0.15 | | | | -0.03 | | | | 0.30 | | | |
| Agreement (Fleiss kappa, per response) | | 0.04 | -0.02 | -0.04 | n/a | 0.17 | 0.12 | 0.15 | n/a | 0.29 | 0.11 | 0.33 | n/a | 0.24 | 0.01 | 0.14 | n/a | -0.00 | 0.11 | 0.08 | n/a | 0.36 | 0.20 | 0.28 | n/a |

*Missing rating or rated as 'not applicable' if no request for radiography was made or no advice given.
FY2, foundation year two; PA, physician associates.

of the ED, but that 'minors' care was often completed independently.

Our reviewers were 40% sensitive, 46% specific on judging the clinician type: 68% (13/19) of the PA records were thought to be of a FY2 doctor-in-training and 60% (9/15) vice versa (kappa score for inter-rater agreement 0.15).

Interviewees also presented other aspects related to clinical adequacy, particularly the PAs' stability in the team. The clinicians' familiarity with the longer standing team member PA/s—in contrast to FY2 doctors-in-training on rotation—was raised repeatedly:

> If there was a junior physician over here, and he said oh, what do you think of this wound, which they do ask us. And I say yeah, it needs suturing. I then have to say, but can you suture or do you want me to suture it?……Because I don't know, and some will say oh no, I can't……I've never sutured before, and some will say oh yeah, that's fine, I'll suture it……Whereas I know with PAs they'll suture their own. Because I know that they've got that skill set. Participant 177 advanced nurse practitioner

### Secondary outcome: patient experience

Patients were positive about the care they had received from the PA, but had not understood what the PA role meant, with two participants believing they had been seen by a doctor and another unsure in the context of multiple ED staff:

> I presumed he was a fully-qualified physician, yes his approach and everything was absolutely 100%. Participant 120 patient

Most of our patient participants were receptive to the role on the grounds that it might speed up care, although they were not without concern for the difference in training from a doctor and the diminishment of a senior medical workforce:

> It's good to have another person, another opinion… but would it not perhaps be better to have another doctor? Participant 083 patient's relative

## DISCUSSION
### Summary of findings

The study presents evidence from three English EDs and has demonstrated no difference in safety or appropriateness between PAs and FY2 doctors-in-training. We report no difference in re-attendance rates. Those patients seen by a PA (within PA working hours 08:00–22:00) had a shorter average length of stay in the ED than those seen by doctors-in-training (24 hours working period). Our review of clinical adequacy found few errors and no difference between PAs and FY2 doctors-in-training. Patients appeared relatively unconcerned with the title of the clinician treating them and thought they had been

treated by a doctor; however, they were keen to know that the employment of PAs would not represent a widespread substitution for doctors in the ED.

### How this study is similar or different from prior studies

We believe this to be the first empirical study of the outcomes of care provided by UK-trained PAs in the ED, and the first internationally to include interview and observation data. Additionally, patients' views do not appear to have been previously gathered at the time of the visit (and qualitatively), although there have been previous questionnaire studies in the USA of patient satisfaction, administered after the visit.[28–30]

We reported few differences in the the practice and processes of care—other than prescribing (which PAs currently cannot do independently in the UK)—between PAs and doctors in their second foundation year of training. Our finding of no difference in the primary outcome (ED re-attendance rate within 7 days) for patients of PAs and FY2 doctors-in-training is consistent with the comparisons of nurse-qualified NPCs and FY2 doctors-in-training on which we based our study design[22 23] and other PA literature from the USA.[16] It should be noted that for patients in the majors section of ED, all assessment and treatment plans by FY2 doctors-in-training and NPCs were reviewed and agreed by a senior clinician. Our participants commented frequently on the transient nature of FY2 doctors-in-training, whose rotation in the ED only last 4 months. In contrast, PAs remained long-term and provided continuity in the team. Their accumulated knowledge of the policies and practices (clinical and otherwise) of the department, the consultants and the hospital was reported to enable operational efficiencies. Similar observations about PAs providing continuity within the medical/surgical team have been made in North America and the Netherlands[31–33] and also for other NPCs.[34]

This study's strengths lie in its mixed-methods approach to the study of PAs in the ED, allowing consideration of different types of data on their contribution, compared with that of FY2 doctors-in-training, to be considered. We were able to carry out a well-powered quantitative comparative analysis of the documented processes and outcomes of patient care by PAs and FY2 doctors-in-training in three EDs in different parts of the country, and to gather qualitative data on PAs 'in practice'. The qualitative component of our mixed-methods approach enabled contextual explanations of the quantitative analysis.

Our study however has several limitations. Our comparison of PAs and doctors working in all areas of the ED introduced the potential for PAs and FY2 doctors-in-training to be attending to patients of different acuity and complexity. We sought to mitigate this by using three different EDs, taking a sample across a 16-week period at all times of day and night (although the FY2 doctors-in-training worked over the 24 hours period when staff:patient ratios may have fluctuated). We also made statistical adjustments that included triage category. The low

sensitivity of most ED triage systems to identification of conditions other than the most serious, however, is a drawback.[35] The prevention of collection of 28-day outcomes by NHS organisations was also a barrier, particularly as we had based our sample size calculation on that, as opposed to the lower 7-day return rate.

The level of missing data for some variables in the routinely collected data, and not having data from which to take into account whether PA reduced the staff:patient ratio (or fully replaced FY2 doctors-in-training) is a further limitation and needs to be borne in mind in the comparisons we present. Likewise, our observation data illustrated care is predominantly delivered by teams which creates difficulties in attributing outcomes or processes to individual staff, and compromised our ability to undertake an economic evaluation.

Our interview invitations yielded relatively small numbers of participants, particularly among patients/relatives. While we attribute this in part to the fast patient throughput of the ED and limited availability of the researcher, this limits our analysis.

### Implications for policy and practice

PAs in the ED are acceptable to patients and can help to relieve staffing pressures and improve efficiency in the delivery of care. They are able to treat patients safely with a range of conditions and FY2 doctors-in-training deliver similar care to that provided by doctors in their second year of training. Deployment of PAs within ED teams is a potential solution to the situation of growing patient demand and predicted shortage of junior doctors in the British NHS,[7] of which FY2 doctors on rotation in specialties such as the ED are one part; it is not our intention to raise or limit PAs to one particular junior doctor comparator level, but we have used this here as the closest pragmatic comparator. An alternative, which is to hire locum doctors, comes at a higher costs and loss of team continuity, and has potential implications for patient safety. Moves to regulate the PA profession under the General Medical Council were started in 2019.[36]

The findings of this study support employment of appropriately trained, supervised PAs with professional registration in ED teams. Further research is needed to investigate fully the impacts we have observed, particularly the cost-effectiveness.

### Author affiliations
[1]Faculty of Health, Social Care and Education, Kingston University and St George's, University of London, London, UK
[2]Centre for Medication Safety and Service Quality, Imperial College Healthcare NHS Trust, London, UK
[3]School of Economics, University of Surrey, Guildford, UK
[4]Institute of Clinical Sciences, University of Birmingham, Birmingham, UK
[5]Nuffield Department of Primary Care Health Sciences, University of Oxford, Oxford, UK
[6]School of Law and Social Science, Royal Holloway University of London, Egham, UK
[7]Chester Medical School, University of Chester, Chester, UK
[8]Royal Orthopaedic Hospital, Birmingham, UK

**Acknowledgements** The authors would like to thank all those clinicians, administrative and information staff in the participating centres who assisted the study at a time of heightened workload within the emergency services in the National Health Service in England. The authors would also like to thank Robert Grant who provided statistical advice in the design of the study and obtained research funding but left the research team before the data were obtained for analysis.

**Contributors** VD (PhD, health policy and service delivery research), MH (PhD, health services research), JP (MD, general practice and clinical education), HG (PhD, health economics), SdeL (MD(Res), general practice and information science), JG (PhD, medical sociology), SB (BSc, patient and public engagement) and PB (PhD, audiology and strategic management) conceived and designed the study and obtained research funding. VD, MH and JP supervised the conduct of the study and data collection. VD, MH and JP undertook recruitment of participating centres and managed the data, including quality. CWa (PhD, statistics) undertook the statistical analysis; CWh (PhD, health services research), LN (PhD, health services research), MH, JE (MSc, physician associate and education) and VD undertook qualitative data collection and thematic analysis and HG considered the economic aspects. MH drafted the manuscript, and all authors contributed substantially to its revision. VD takes responsibility for the paper as a whole. All authors attest to meeting the four ICMJE.org authorship criteria: (1) substantial contributions to the conception or design of the work; or the acquisition, analysis or interpretation of data for the work; (2) drafting the work or revising it critically for important intellectual content; (3) final approval of the version to be published and (4) agreement to be accountable for all aspects of the work in ensuring that questions related to the accuracy or integrity of any part of the work are appropriately investigated and resolved.

**Funding** This project was funded by the National Institute for Health Research Health Services and Delivery Research Programme (project number 14/19/26). This paper presents independent research commissioned by the National Institute for Health Research (NIHR). The protocol for the study is available at the website https://www.journalslibrary.nihr.ac.uk/programmes/hsdr/141926/#/.

**Disclaimer** The views and opinions expressed by authors in this publication are those of the authors and do not necessarily reflect those of the NHS, the NIHR, the Health Service and Delivery Research Programme or the Department of Health.

**Competing interests** SdeL was head of the Department of Clinical and Experimental Medicine until June 2019 at the University of Surrey, which launched a physician associate course in 2016. JP is the immediate past chair of the UK and Ireland Board for Physician Associate Education and immediate past director of the physician associate programme at the University of Birmingham. PB is honorary faculty at the University of Birmingham and has taught on the physician associate programme since 2008. JE taught part time on the University of Birmingham physician associate programme until 2020. VMD was a HS&DR Board Member in 2015.

**Patient and public involvement** Patients and/or the public were involved in the design, conduct, reporting or dissemination plans of this research. Refer to the 'Methods' section for further details.

**Patient consent for publication** Not required.

**Ethics approval** This study was approved by the NHS Health Research Authority London-Central Research Ethics Committee (15/LO/1339).

**Provenance and peer review** Not commissioned; externally peer reviewed.

**Data availability statement** No additional data are available for sharing.

**ORCID iDs**
Mary Halter http://orcid.org/0000-0001-6636-0621
Vari Drennan http://orcid.org/0000-0002-8915-5185
Simon de Lusignan http://orcid.org/0000-0001-5613-6810

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
