## [Reviewer comments · BMJ Open]

ARTICLE DETAILS

TITLE (PROVISIONAL)	Comparing physician associates and foundation year two doctors in training undertaking emergency medicine consultations in England: a mixed methods study of processes and outcomes
AUTHORS	Halter, Mary; Drennan, Vari; Wang, Chao; Wheeler, Carly; Gage, Heather; Nice, Laura; de Lusignan, Simon; Gabe, Jonathan; Brearley, Sally; Ennis, James; Begg, Phil; Parle, Jim

VERSION 1 – REVIEW

REVIEWER	Markus Bleckwenn Department of General Practice Faculty of Medicine, University of Leipzig Ph.-Rosenthal-Str. 55 04103 Leipzig Germany
REVIEW RETURNED	08-Mar-2020

GENERAL COMMENTS	The methods section should describe in more detail how the interview partners were selected, random selection? Why did you interview "only" 6 patients and/or relatives? Was there already a saturation of the answers? I have no questions about the quantitative part of the study. Overall, an important study to reduce the workload of the emergency rooms.
--

REVIEWER	Mr Neil Howie PGCert PGDip MSc FHEA PA-R Course Director, MSc Physician Associate Programme, Department of Paramedic Science and Physician Associates, School of Allied Health and Community, University of Worcester I am a Senior Lecturer on the MSc Physician Associate course at the University of Worcester and the course director since November 2017. I am a Physician Associate and worked in Emergency Medicine from 2011 to 2013. A paper that I authored has been cited in this study (Howie, 2015).
REVIEW RETURNED	14-Apr-2020

GENERAL COMMENTS	This study has used a very logical and well set out methodology which can be reproduced elsewhere due to the clarity with which it is discussed. The discussion of strengths and weaknesses is strong and helpful. One note, the decision to chose FY2 doctors as the comparator to PAs is a little vague, even after reading the associated citation. I was unsure if my understanding of this comparison was more from
--

	my own experience as a PA in ED or from the citation. The term "junior doctor" is relatively broad and some more detail would have been useful, as this does generate a bench-mark of PA = FY2 that may be limiting in future workforce development of development of individuals in their careers.
--	---

REVIEWER	Marianne Lisby Research Center for Emergency Medicine, Aarhus University Hospital, Palle Juul-Jensens Boulevard 161, 8200 Aarhus N, Denmark Department of Clinical Medicine, Aarhus University, Palle Juul-Jensens Boulevard, 8200 Aarhus N, Denmark
REVIEW RETURNED	04-May-2020

GENERAL COMMENTS	This is an interesting and well-written manuscript. The study investigates the contribution of physician associates (PAs) compared to foundation year two doctors (FY2 doctors) in training in terms of re-attendance within 7 days using a mixed-methods approach. Moreover, it investigates the consultation process and the clinical adequacy of care, and how the staff, patients or relatives experienced the PAs. The authors found that the PAs performed equally safe and appropriate to that of FY2 doctors. The manuscript has some major and minor issues that need to be addressed further. Major issues 1) What is the correct study population in the manuscript? In the abstract (methods) 3197 is mentioned, however in table 1-3 the denominator is stated as 8816, and the proportions are calculated based on 8816 records, despite the primary outcome "re-attendance" only was available for 3197 patients. The huge number of "unknown" in the data set (almost 2/3) makes it quite confusing to correctly interpret the results. What was the reason for these "missing/unknown data" and how does that influence the trustworthiness of the findings? How come you didn't limit the study population to records including the primary outcome? As an example - in the abstract you write that the occurrence of re-attendance was 2.2% (N=194) and the sample size was stated as 3197 patients (participants). Re-calculating these numbers showed a prevalence of $194/3197=6.06\%$ - which is 3 times more. I would recommend that you carefully go through the presentation of your data and decide whether the denominator is 3197 or 8816? In case of the latter, you must carefully explain the huge number of the missing data and the impact on the findings. 2) The number of PAs was n=6 and the number of FY2 doctors were n=22. This means that data may not be independent as each of them was involved in large number of the patients. In addition, the PAs may have been involved in 3-4 times more of the included patients than the FY2 doctors. If one or more of them performed extremely well or bad this may influence the results. Did you check for dependency among the participants in the two groups? And if yes, what was the outcome of this test. If not I would recommend to do so. 3) It is not clear from the manuscript whether patients could be included more than once in the chart review sample? If they were included more times during the 16 weeks data collection period this may also have created a dependency among data from the patients. Please clarify how the sample was collected and how you have/will compensate for possible dependency among patients. 4) Table 4 is not self explanatory and difficult to follow. In the main text it is stated that 37/40 cases was appropriate; however this is
---

	not easy to capture from the table. Please explain the results in more details e.g. how many times did the reviewers disagree on the single components in the clinical adequacy assessment. The low Kappa values suggest a high level of disagreement between the reviewers,;however, it would be more reader-friendly to have access to exact numbers. Moreover, the Kappa statistics is not described in the method section (Analysis, page 10). If it was used as a measure of agreement between the four reviewers, did you use the ordinary Kappa statistics or Fleiss Kappa intended for two or more raters? Please, describe. Minor issues: Abstract: - P3, L 34-35: Please specify whether the study population was 3197 or 8816 – se comment 1) What this study adds: - P6, L31: I am not sure that your study findings warrants the following statement "...more broadly for regulation of the profession [PA]; though, it may be a possible spinoff. Methods: P9, L10-16: The description of the sample-size calculation is not clear. How did you arrive at the estimates 18.3% and 27.4%, when the national average was 7.4% (2.4-21.7%)? P10, L20-36: Please describe the statistical analysis for agreement between the reviewers (Kappa / Fleiss Kappa) – se comment 4) P12, L4-5 (Ethical approval): There is no description of how consent was achieved from the interviewed patients, relatives and staff. Please describe. Discussion: - P18, L26: Here you mention 28 days re-attendance – shouldn't it be 7 days re-attendance? - P18+19 (limitations): This section should be elaborated further in terms of dependency among data and the huge amount of "unknown/missing's". Se comment 1-2) above.
--	---

VERSION 1 – AUTHOR RESPONSE

Reviewer(s)' Comments to Author:

Reviewer: 1

Reviewer Name: Markus Bleckwenn

Thank you for your review; we answer each of your points below:

1. The methods section should describe in more detail how the interview partners were selected, random selection?

We have added a sentence to the methods section, detailing how interview participants were invited via the researcher on site or by their site manager – while we had purposive sampling criteria, our sample within those criteria was convenience.

See page 9, lines 4-7

2. Why did you interview "only" 6 patients and/or relatives? Was there already a saturation of the answers?

We only interviewed this number of patients/relatives due to the fast paced ED environment in which we did not invite the patient to interview until the patient had been assessed and usually treated by

the physician associate and the patient was ready to be discharged to their next destination, and this within the availability of the researcher during observation periods.
We now refer to this in the limitations section, page 20, lines 13-15.

3. I have no questions about the quantitative part of the study. Overall, an important study to reduce the workload of the emergency rooms.
Thank you for your positive comment on our study.

Reviewer: 2

Reviewer Name

Mr Neil Howie PGCert PGDip MSc FHEA PA-R

Thank you for your review; we answer each of your points below:

1. This study has used a very logical and well set out methodology which can be reproduced elsewhere due to the clarity with which it is discussed. The discussion of strengths and weaknesses is strong and helpful.

Thank you for your positive comments on the paper; you will note that some requests have been made for further clarity from reviewers, which we have sought to address.

2. One note, the decision to choose FY2 doctors as the comparator to PAs is a little vague, even after reading the associated citation. I was unsure if my understanding of this comparison was more from my own experience as a PA in ED or from the citation. The term "junior doctor" is relatively broad and some more detail would have been useful, as this does generate a bench-mark of PA = FY2 that may be limiting in future workforce development of development of individuals in their careers.

Thank you for raising this omission in the paper; we apologise as this detail was in earlier iterations which we have perhaps lost to word reductions. We have now (re)inserted our rationale for selecting this comparator in the methods section (page 7, lines 18-20), and added to the discussion that this is not intended to be a limiter, rather was a pragmatic decision on the basis of advice that this – at the time of the study – was the 'closest relative' to many PAs in terms of the medical day-to-day activity (page 20, line 23-26).

Reviewer: 3

Reviewer Name

Marianne Lisby

Thank you for your review; we answer each of your points below:

1. This is an interesting and well-written manuscript.
Thank you for your positive view of our manuscript overall; we aim below to address the major and minor issues that you consider need to be addressed further.

2. Major issues

1) What is the correct study population in the manuscript? In the abstract (methods) 3197 is mentioned, however in table 1-3 the denominator is stated as 8816, and the proportions are calculated based on 8816 records, despite the primary outcome "re-attendance" only was available for 3197 patients. The huge number of "unknown" in the data set (almost 2/3) makes it quite confusing to correctly interpret the results. What was the reason for these "missing/unknown data" and how does that influence the trustworthiness of the findings? How come you didn't limit the study population to records including the primary outcome?

As an example - in the abstract you write that the occurrence of re-attendance was 2.2% (N=194) and the sample size was stated as 3197 patients (participants). Re-calculating these numbers showed a prevalence of $194/3197=6.06\%$ - which is 3 times more. I would recommend that you carefully go through the presentation of your data and decide whether the denominator is 3197 or 8816? In case of the latter, you must carefully explain the huge number of the missing data and the impact on the findings.

Thank you for pointing out our inconsistency here, which we had not intended. The confusion arises as our sample for some of the process measures comes from the original n=8816 cases and for this we wished to use the largest sample we had available. However, the primary outcome was not routinely available in two of our sites, and had to be collected manually – this was only agreed for the subsample upon which our sample size calculation was based.

We have made a decision to keep in the larger sample, as the process outcomes have important implications for practice, and we are better powered for these secondary analyses with the inclusion of all the cases for which we have data. You are, however, correct, that the denominator for the proportion we present for the primary outcome of re-attendance should have been the number for which we have the primary outcome, i.e. 3197. This is closer to the 8% we reported for our analysis of a smaller sample in our main study report.

We have now corrected this throughout the manuscript (abstract page 3, lines 17-18; results p10, lines 12-14 and Table 1) and have explained that the missing data occur as they were not routinely electronically available for the study team, not that they were truly missing in the patients' records.

2) The number of PAs was n=6 and the number of FY2 doctors was n=22. This means that data may not be independent as each of them was involved in large number of the patients. In addition, the PAs may have been involved in 3-4 times more of the included patients than the FY2 doctors. If one or more of them performed extremely well or bad this may influence the results. Did you check for dependency among the participants in the two groups? And if yes, what was the outcome of this test. If not I would recommend to do so.

We thank the reviewer for suggesting this additional analysis. We have calculated intraclass correlation coefficients (ICC, which measures the correlation between the outcomes of patients by the same clinician) for each outcome and report results using a random-effects model if the ICC is statistically significant. For the primary outcome, i.e. re-attendance, the ICC is small (0.008) and statistically insignificant ($p=0.236$) so the original model specification was used. We mention this now in the methods (page 9, lines 18-20).

For the secondary outcomes, however, the ICC was found to be significant for three outcomes, and we have re-run the models to also include this clustering in the data. As the reviewer suspected, this has impacted on our results, with a reduction in the OR (but remaining statistically significant) for prescriptions received, and one where a previously statistically significant result now presents as non-significant – admission from the ED. We have changed the results in tables 2 and 3, in the results narrative (page 13, lines 16-21), in the discussion (page 18, line 31) and in the abstract (page 3, lines 28-30).

In order to undertake this analysis we imported staff identifiers from our original data file supplied by the NHS organisations into our fully anonymised analysis data file. During this process we identified an error in the profession variable, that is, whether the attending clinician was a PA or doctor in one set of cases – for one site, in the cases that were only used in the secondary analysis where additional data were available, and in one case used for all analyses – where the PA/FY2 label had been transposed. This has resulted in changes to Table 1 where we report the sample characteristics by PA/FY2, but had made only very minor difference to the presented analyses.

3) It is not clear from the manuscript whether patients could be included more than once in the chart review sample? If they were included more times during the 16 weeks data collection period this may also have created a dependency among data from the patients. Please clarify how the sample was collected and how you have/will compensate for possible dependency among patients.

The sample was collected by NHS trust staff and the conditions for the information release included that it was provided anonymously to the research team; on account of this it was not possible to assess whether patients were included more than once in the sample. We have now emphasised this in the methods where we already referred to a pseudo-anonymised sample (page 7, line 18 and page 8, line 2), and we state that not being able to assess dependency of the data is a limitation.

4) Table 4 is not self explanatory and difficult to follow. In the main text it is stated that 37/40 cases was appropriate; however this is not easy to capture from the table. Please explain the results in more details e.g. how many times did the reviewers disagree on the single components in the clinical adequacy assessment. The low Kappa values suggest a high level of disagreement between the reviewers; however, it would be more reader-friendly to have access to exact numbers. Moreover, the Kappa statistics is not described in the method section (Analysis, page 10). If it was used as a measure of agreement between the four reviewers, did you use the ordinary Kappa statistics or Fleiss Kappa intended for two or more raters? Please, describe.

On reflection we can now see that the table versus the text was not entirely clear. We have added a simple explanation of the two column headings that are added together to form our statement on the number of appropriate cases and hope this makes this clear (page 16, lines 21-22).

We used Fleiss' kappa and have now placed that in the methods section, page 9, lines 26-27. This was conducted for each of the six components of the assessment of clinical adequacy, as shown in Table 4.

Minor issues:

Abstract:

- P3, L 34-35: Please specify whether the study population was 3197 or 8816 – see comment 1) What this study adds:

We have amended the study population figures, as described above, and removed the 'What this study adds' section

- P6, L31: I am not sure that your study findings warrants the following statement "...more broadly for regulation of the profession [PA]; though, it may be a possible spinoff.

The editor has requested that this section of the paper is no longer required, so this statement is now removed.

Methods:

P9, L10-16: The description of the sample-size calculation is not clear. How did you arrive at the estimates 18.3% and 27.4%, when the national average was 7.4% (2.4-21.7%)?

The sample size was arrived at as stated by using the references cited on comparison of nurse practitioners and junior doctors in the ED. At that point in time the study intended to collect 28 day outcome data – this was explicitly prevented by the sites who declined access to the data for researchers or a research nurse to send surveys to patients. We therefore became dependent on the hospital-collected 7 day return outcome. We have added a brief explanation of this to the paper (page 9, lines 8-14). We have added this to the limitations (page 20, lines 2-6).

P10, L20-36: Please describe the statistical analysis for agreement between the reviewers (Kappa / Fleiss Kappa) – see comment 4)

Please see above

P12, L4-5 (Ethical approval): There is no description of how consent was achieved from the interviewed patients, relatives and staff. Please describe.

We have added this in the methods section.

Discussion:

- P18, L26: Here you mention 28 days re-attendance – shouldn't it be 7 days re-attendance?

Yes, our measure was 7 days; the confusion arises from the fact that the papers cited measured 28 days. We have now sought to explain this by adding two sentences explaining why we had to move from a prospective design that included 28 day follow up (hence our sample size calculation) to retrospective routine data use only with a 7 day return rate. We trust this now explains this more clearly.

- P18+19 (limitations): This section should be elaborated further in terms of dependency among data and the huge amount of "unknown/missing's". See comment 1-2) above.

We had mentioned the missing data (page 20, line 9), and hope we have addressed the dependency in the data in the results section.

VERSION 2 – REVIEW

REVIEWER	Markus Bleckwenn Department of General Practice, Faculty of Medicine, University of Leipzig, Germany
REVIEW RETURNED	19-Jun-2020

GENERAL COMMENTS	P 56 line 3: 194 out of 3197 is 6.1%
--------------------------------------

REVIEWER	Marianne Lisby Research Center for Emergency Medicine, Aarhus University, Department of Clinical Medicine Denmark
REVIEW RETURNED	18-Jun-2020

GENERAL COMMENTS	Thanks for a thoroughly revised manuscript. The authors have sufficiently addressed the issues brought forward in the previous review; however, in relation to question 4 (Kappa Fleiss) the authors should consider to add reporting of "per response" inter-rater agreement (Analysis section).
--

VERSION 2 – AUTHOR RESPONSE

Reviewer 3

The authors have sufficiently addressed the issues brought forward in the previous review; however, in relation to question 4 (Kappa Fleiss) the authors should consider to add reporting of "per response" inter-rater agreement (Analysis section).

We thank the reviewer for her positive comment, and note that we have added to the methods section that we have conducted a per response kappa score (page 9, line 29), and changed the word 'by' to 'per response' in the final row of table 4, page 37, for consistency.

Reviewer 1

P 56 line 3: 194 out of 3197 is 6.1%

Thank you for noticing our oversight in not correcting the % in this place in the manuscript; the correction is now at line 3, page 13.